# Validation of Multiplex PCR and Serology Detecting *Helicobacter* Species in Mice

**DOI:** 10.3390/microorganisms11020249

**Published:** 2023-01-18

**Authors:** Julia Butt, Mareike Schmitz, Bernhard Berkus, Katja Schmidt, Daniela Höfler

**Affiliations:** 1Infections and Cancer Epidemiology, German Cancer Research Center (DKFZ), 69120 Heidelberg, Germany; 2Microbiological Diagnostics, German Cancer Research Center (DKFZ), 69120 Heidelberg, Germany

**Keywords:** *Helicobacter*, routine health monitoring, multiplex serology, multiplex DNA, laboratory mice

## Abstract

High-throughput multiplexed assays are needed to simplify detection of *Helicobacter* species in experimental infection and routine health monitoring of laboratory mice. Therefore, fluorescent bead-based hybridization assays for *Helicobacter* sp. DNA and serology were developed. Multiplex PCR amplicons (*H. hepaticus*, *H. bilis, H. typhlonius*, *H. pylori*, *H. muridarum*, *H. pullorum*, *H. cinaedi*, *H. heilmanii*, *C. jejuni*) and antibodies against *H. pylori*, *H. hepaticus*, *H. bilis* were assessed in naturally and experimentally infected mice, and results compared to conventional PCR. Species-specific and sensitive detection of seven *Helicobacter* spp. <100 copies/PCR, and of two species <1000 copies/PCR was successfully established in the *Helicobacter* multiplex DNA finder. The novel assay was highly comparable with conventional PCR (kappa = 0.98, 95%CI: 0.94–1.00). Antibody detection of *H. hepaticus* and *H. bilis* showed low sensitivity (71% and 62%, respectively) and cross-reactivity in *H. typhlonius*-infected mice. Infection experiments showed that antibodies develop earliest two weeks after DNA detection in feces. In conclusion, detection of *Helicobacter* antibodies showed low sensitivity depending on the timing relative to infection. However, *Helicobacter* multiplex DNA finder is a sensitive and specific high-throughput assay applicable in routine health monitoring for laboratory animals.

## 1. Introduction

The genus *Helicobacter* includes over 30 formally assigned species infecting a variety of hosts, including rodents and humans. The genus is subdivided into two groups, enterohepatic and gastric *Helicobacter* spp., depending on their properties to colonize the respective organs [1].

The species *Helicobacter (H.) hepaticus* and *H. bilis*, for example, belong to the enterohepatic group and were both first identified in mice [1,2,3]. Natural and experimental infections in mice are associated with inflammatory and cancerous diseases of the enterohepatic tract [4,5,6,7,8,9,10,11,12,13,14]. Furthermore, an infection of laboratory mice with *H. hepaticus* and *H. bilis* has been shown to alter experimental outcomes including animal research on *H. pylori*, a human carcinogen for the development of gastric cancer [1]. As recommended by the Federation of European Laboratory Animal Science Associations (FELASA), routine monitoring of these infections is crucial to microbiologically standardize mice used in animal experiments, not only to reduce the number of animals used in experiments (in line with the 3R principle—replacement, reduction and refinement) but also to avoid misinterpretation of data [15]. Besides these infections, FELASA also recommends the detection of *H. typhlonius*. Thus, there is a need for sensitive, specific and at best high-throughput techniques to detect the presence of *Helicobacter* at the genus and species levels, to differentiate at least *H. hepaticus*, *H. bilis*, and *H. typhlonius* for health monitoring in animal facilities of research institutions.

Diagnostic tests applied to monitor *Helicobacter* infections in animal facilities commonly use fecal samples for detection of DNA. Although this is usually realized by conventional or quantitative PCR, available diagnostic methods often have shortcomings in their specificity [16]. PCR testing for detection of *Helicobacter* DNA is usually based on the highly conserved 16S rRNA sequence. However, this sequence shows on average a 94% homology on the nucleic acid sequence level among the *Helicobacter* spp. mentioned above as well as other *Helicobacter* spp. that infect rodents and/or humans, i.e., *H. pylori*, *H. muridarum*, *H. heilmanii*, *H. cinaedi*, and *H. pullorum* [2,17,18,19,20]. This low specificity often allows the differentiation of a limited number of *Helicobacter* spp. only. Moreover, singleplex PCR analyses as well as analyses using gel electrophoresis are time- and labor-intensive and less suited for high-throughput testing.

Apart from molecular methods, serological testing for the detection of infectious agents to monitor current and past infections is used for health monitoring. The presence of antibodies to these infectious agents, primarily against rodent-associated viruses, is usually analyzed in serum obtained from sentinel mice. Multiplex serology approaches would also allow incorporating *Helicobacter* serology into these routinely performed diagnostics and could therefore supplement PCR testing to potentially reduce labor and costs. The majority of *H. bilis* and *H. hepaticus* serological assays are based on whole bacteria or membrane protein extracts with a high potential of detecting cross-reactive antibody responses resulting from reactions to other *Helicobacter* spp. [21,22,23]. Attempts to use individual proteins in ELISA identified sensitive and specific sero-responses to proteins P167C and D for *H. bilis* and HH0435 for *H. hepaticus*, providing potential candidates for development of a multiplex serology assay [24,25,26].

The aim of this project was to develop a fluorescent bead-based species-specific multiplex PCR for the detection of *Helicobacter* spp., including *H. hepaticus*, *H. bilis*, and *H. typhlonius*, and also of closely related rodent- and/or human-infecting *H. pylori*, *H. heilmannii*, *H. cinaedi*, *H. pullorum*, and *H. muridarum* in murine fecal samples to cover not only FELASA-recommended species for health monitoring in animal facilities but also other species potentially relevant for research purposes. We furthermore included one distantly related *Campylobacter* species (*C. jejuni*) as a specificity control. The newly developed multiplex PCR was compared to conventional PCR to assess sensitivity and specificity. We further assessed the corresponding antibody response in mice by using newly developed *H. bilis* and *H. hepaticus* antigens incorporated in an already existing multiplex serology for *H. pylori* [27]. To assess more closely the natural history of antibody development to *H. bilis* and *H. hepaticus* in mice, we also performed an infection experiment described here.

## 2. Materials and Methods

### 2.1. Origin, Housing and Sampling of Mice

All mice were housed at the vivarium of the German Cancer Research Center (DKFZ), Heidelberg. The animal facility of the DKFZ has been officially approved by the responsible authority (Regional Council of Karlsruhe, Karlsruhe, Germany) under the official approval file no Az 35-9185.64BH DKFZ. Housing conditions are thus in accordance with the German Animal Welfare Act (TierSchG) and the EU Directive 2010/63/EU [28]. Compliance with institutional guidelines and legal regulation regarding care and handling of animals was ensured by designated veterinarians according to article 25 of Directive 2010/63/EU and by the Animal Welfare Body according to article 27 of Directive 2010/63/EU [28].

To assess sensitivity and specificity of multiplex PCR and serology, we used samples of n = 340 colony animals and sentinel mice and mastomys (different transgenic strains, CD1, C57BL/6, NMRI and Mastomys coucha), which were tested in the course of routine health monitoring of rodents at the DKFZ. The samples for PCR analysis comprised duodenal and liver samples from n = 16 mice, as well as n = 209 fecal samples (n = 109 from individual mice and n = 100 pooled fecal samples from two to three mice housed in the same cage). Concurrent serum samples were available from n = 338 mice (Appendix A). Analyzing animal samples for the detection of infections in the course of health monitoring at the DKFZ did not require official approval by the local governmental authorities.

### 2.2. Helicobacter Infection Experiment, Housing and Handling of Mice

For the infection experiment, CD1 (official strain nomenclature Crl:CD1 (ICR); Crl strain code 022) mice were bred for biotechnical and health monitoring purposes under high hygiene conditions as gnotobiotic mice, colonized with the Taconic Altered Schaedler flora. Mice were housed in individually ventilated cages (IVC) (GM500, Greenline, Tecniplast, Buguggiate, Italy). All cage beddings (aspen material), nesting material (aspen wood, 24–120 mm, Abedd Vertriebs GmbH, Vienna, Austria), food (Mouse Maintenance No. 3437, KLIBA NAFAG, Kaiseraugst, Switzerland) and water were autoclaved before use. Cage changing was carried out under a laminar flow hood. For sampling, animals were handled in a biosafety 2 cabinet. New overgloves were used after all mice of the same group were handled, and Wofasteril was used for disinfection after working processes. To identify *Helicobacter*-positive mice as donor mice, fecal material was collected and analyzed by conventional PCR. Groups of five CD1 mice were co-housed with *Helicobacter*-positive donor mice and a group of three mice was co-housed with a *Helicobacter*-negative mouse (negative control group). The initial protocol was to co-house each of the different mouse groups with a *H. hepaticus*, *H. typhlonius*, and a *H. bilis* single-infected mouse. However, single-infected donor mice were not available for *H. bilis* consequently the following infection groups were set up: (A) *H. hepaticus* single-infected, (B) *H. typhlonius* single-infected, (C) *H. typhlonius* and *H. hepaticus* dual-infected, and (D) a *H. bilis* and *H. typhlonius* dual-infected group. Before starting the co-housing, feces and blood were taken from the experimental mice and samples were used as reference values (0 weeks post exposure (wpe)). Every second wpe, feces and blood samples was collected and analyzed by multiplex PCR and multiplex serologic assay for a duration of 16 weeks. The infection experiment was officially approved by the local governmental authorities (Regional Council of Karlsruhe, Karlsruhe, Germany) under the notification number G-16/17.

### 2.3. DNA Preparation

DNA was extracted using the Maxwell 16Lev device and the Maxwell 16 FFPE Plus LEV DNA Purification Kit (Promega GmbH, Mannheim, Germany). To a fecal pellet (~40 mg), 300 µL of homogenization solution was added. The sample was mechanically homogenized by a sample homogenizer (Precellys24, Bertin Instruments, Montigny-le-Bretonneux, France) at 5000× *g* for 15 s. Next, 25 µL of the homogenate was added to 200 µL lysis buffer before the mixture was transferred into the first well of the Maxwell processing cartridge. The manufacturer’s instructions were followed for the subsequent steps and the DNA was eluted in 150 µL DNase/RNase free water.

### 2.4. Conventional PCR

One microliter template DNA was used for the PCR reaction. The PCR was run using the Promega GoTaq^®^ G2 Hot Start Polymerase kit (Promega GmbH, Mannheim, Germany) according to the manufacturer’s instructions with a total volume of 25 µL. The primers used for the genus- and species-specific *Helicobacter* PCR are given in Table 1. The reaction conditions and the appropriate annealing temperatures for primers (53 °C for *Helicobacter* genus and *H. typhlonius*, 56 °C for *H. bilis*, 61 °C for *H. hepaticus*) were adjusted according to the manual supplied by Promega with 32 (*H. bilis*) or 43 (all others) cycles. Amplicons were analyzed by ethidium bromide gel electrophoresis.

### 2.5. Helicobacter Multiplex DNA Finder

One multiplex PCR amplified eight *Helicobacter* and one *Campylobacter* spp. (*H. muridarum, H. pylori, H. bilis, H. heilmannii, H. typhlonius, H. cinaedi, H. hepaticus, H. pullorum, C. jejuni*). Primers targeted the gyrase A gene and gamma-glutamyltransferase if present. Primers and probes were designed using the LightCycler Probe Design Software 2.0 (Roche). To ensure high clinical specificity, forward or backward primers or probes were designed with as many mismatches as possible to sequences of related species using the National Center for Biotechnology Information (NCBI) BLAST database. All primers were assessed for primer-dimer formation and for unspecific annealing of biotinylated primers to oligonucleotide probes by Thermofisher Multiple Primer Analyzer.

The multiplex PCR was performed in a final reaction volume of 12.5 µL comprising 1× Multiplex PCR Kit buffer (Qiagen, Hilden, Germany), containing 3 mM MgCl_2_, dNTP mix, 0.5× Q-solution and HotStartTaq DNA polymerase, 0.2 to 0.4 µM of each primer (Table 2), and 1 µL of purified DNA. The reaction conditions were run as described earlier but using 40 cycles of amplification in a Mastercycler (Eppendorf, Hamburg Germany) [29]. The detection of amplicons was performed via hybridization reaction, adding 10 µL of PCR product to the bead mixture containing 33 µL of tetramethylammonium chloride (TMAC) hybridization solution (0.15 M TMAC, 75 mM Tris–HCl, 6 mM ethylen diamin tetraacetate (EDTA), 1.5 g/L Sarkosyl, pH 8), 7 µL of 1× TE and a mixture of 2000 probe-coupled beads. Next, 10 min heat denaturation at 95 °C, 30 min hybridization at 41 °C under stringent conditions, and 20 min incubation with streptavidin-R-phycoerythrin (Roche Diagnostics, Mannheim), followed by Luminex read-out resulted in median fluorescence intensity (MFI) values/target for each specimen as described in detail earlier [30].

### 2.6. Helicobacter Multiplex DNA Finder Controls

As positive control, plasmid-DNA was extracted from a dam^+^, dcm^+^
*E. coli* strain containing the selected bacterial target sequences (Eurofins, Ebersberg, Germany) as described before [29]. The copy number/unit mass was calculated by assuming that 1 bp weighs about 660 Da. Concentration of plasmid-DNA was measured with the NanoDrop 1000. Knowledge of the concentration of the purified DNA preparations allowed computing the number of plasmid/µL that was used to determine the analytical sensitivity of *Helicobacter* multiplex DNA finder. For internal quality control of mouse DNA, polymerase A gene was co-detected in the *Helicobacter* multiplex DNA finder.

In all multiplex PCR and hybridization runs, reactions without template DNA were used as assay negative controls indicating reagent contamination.

To check the assay’s specificity, whole genomes of three closely related bacteria (*H. canis* (ATCC 51402)*, C. lari* (DSM 11375-0313-001)*, C. coli* (ATCC 4994)) as well as Helicobacter-negative murine fecal samples were applied to the *Helicobacter* multiplex DNA finder.

### 2.7. Cut-Off Definition of Helicobacter Multiplex DNA Finder

For each probe, MFI values in reactions with no PCR product added to the hybridization mixture were considered as background values. Net MFI values were computed by subtraction of 1.2 times the maximal background value plus 5 MFI [29]. All samples were applied in duplicates. Samples were defined as *Helicobacter*/*Campylobacter* positive if the net MFI values in both duplicates were above the cut-off net value of 1 or the net MFI value in one duplicate at least above 5.

A sample was defined as valid if either *Helicobacter*/*Campylobacter* spp. was positive or the polA control was positive with a net MFI value above the cut-off net value of 1 or the net MFI value in one duplicate at least above 5.

### 2.8. Helicobacter Multiplex Serology

Antibody responses to n = 13 *H. pylori* proteins were determined as described previously [27]. For the analysis of antibody responses to *H. bilis* and *H. hepaticus,* we selected each seven potential antigens either based on previously published literature regarding immunogenicity (P167C, P167D [24] and HH0435 [26]) and function as potential virulence factor (HRAG01818 [31], HH0243 [32], and HH1446 [31,32]), or as homologues to known immunogenic proteins of *H. pylori* (HRAG1504 and HH0713 to HP1564; HRAG00845 and HH1201 to GroEl; HRAG1407 and HH0407 to UreA; HRAG01298 and HH0966 to HP0305, respectively) [27]. We determined the amino acid sequence identity to proteins of other closely related *Helicobacter* spp. using BlastP to assess potential cross-reactive antibody responses (Appendix A) [33]. All selected proteins were recombinantly expressed as Glutathione-S-Transferase (GST)-tag fusion proteins in *E. coli* BL21 as described previously and applied in the multiplex serology assay [34]. Briefly, GST-tag fusion proteins were affinity-purified on fluorescently labelled polystyrene beads (Luminex Corp., Austin, TX, USA). Mixing of the differently labelled bead sets resulted in a suspension array that was incubated with serum (final serum dilution 1:100). The quantity of bound antibodies was detected by biotinylated goat anti-mouse IgG secondary antibody (Dianova, Hamburg, Germany) and a reporter fluorescence, streptavidin-R-phycoerythrin (Moss Inc., Pasadena, MD, USA). A Luminex 200 analyzer (Luminex Corp., Austin, TX, USA) then distinguished the bead set as well as quantified the amount of bound serum antibody as the median reporter fluorescence intensity (MFI) of at least 100 beads/bead set/serum samples. Cut-offs for antigen-specific sero-positivity were defined to allow for a maximum of 5% sero-positives among *Helicobacter* genus DNA-negative mice.

### 2.9. Statistical Analysis

The agreement of conventional PCR and the *Helicobacter* multiplex DNA finder was indicated by chi-square test of independence and by kappa statistics (k), where a value of one represents complete, zero represents no agreement.

Continuous MFI values obtained in multiplex serology assays were compared between groups by Wilcoxon–Mann–Whitney test. A *p*-value < 0.05 was considered statistically significant.

All graphical presentations and statistical analyses were carried out using GraphPad Prism 8 (San Diego, CA, USA).

## 3. Results

### 3.1. Analytical Sensitivity of Helicobacter Multiplex DNA Finder

Ten-fold dilution series of plasmid DNA containing the selected target sequences of eight *Helicobacter* and one *Campylobacter* species plus a genus-specific *Helicobacter* sequence were applied to *Helicobacter* multiplex DNA finder resulting in an analytical sensitivity below 100 copies/PCR for all *Helicobacter*/*Campylobacter* spp. but 1000 copies/PCR for H. *typhlonius* and H. *heilmannii* and 10,000 for *C. jejuni* when targeting the gyraseA gene (Table 3). Despite the presence of 50 ng/µL mouse DNA, analytical sensitivity remained as described above, but decreased 10-fold for *H. muridarum* and *C. jejuni* targeting the ggt gene. The detection of the murine polA gene DNA quality control reached the level of 10 copies/PCR corresponding to about five cell equivalents. To assess the robustness of the *Helicobacter* multiplex DNA finder, the same 10-fold dilution series were analyzed in duplicates on the same plate and in two individual experiments on two different days. Of 60 expected signals at the detection limit (four expected signals in duplicates on two different days * targeted gene), all could be detected, indicating a high reproducibility.

### 3.2. Specificity of Helicobacter Multiplex DNA Finder

Specificity was analyzed by applying 10^6^ plasmid copies/PCR that contained the selected target sequences of nine *Helicobacter* and *Campylobacter* spp. plus one genus-specific *Helicobacter* sequence to the *Helicobacter* multiplex DNA finder. Additionally, specificity of polA gene detection was tested by applying 50 ng of fecal DNA. Detection of all *Helicobacter* and *Campylobacter* spp. (one gene target for each of the nine species and one additional gene target for four species, plus one for the genus *Helicobacter*) and the polA-specific quality control was highly specific. Only *H. cinaedi* and *H. typhlonius*, *H. cinaedi* and *H. pylori* showed weak expected cross-reactivities (below 10% of the specific MFI signal) due to the high homology of probe and primer sequences, with only four to six mismatches in their nucleotide sequence (Table 4). The unexpected cross-reactivities of the polA gene with *H. pullorum* and *H. typhlonius* probe were also below 10% of the specific polA MFI signal. Whole genomes (50 ng/µL) of *H. canis*, *C. lari*, and *C. coli* were not detected in the *Helicobacter* multiplex DNA finder, although sequence homology of the selected *Helicobacter* and Campylobacter sequences were between 88 and 90%. 

### 3.3. Comparison of Helicobacter Multiplex DNA Finder to Conventional PCR

For a direct comparison of the conventional PCR and the novel *Helicobacter* multiplex DNA finder, the detection of *H. hepaticus*, *H. typhlonius*, *H. bilis* and of the *Helicobacter* genus were analyzed. Since *H. typhlonius* showed weak cross-reactivities with polA, *H. typhlonius* was defined as positive where the polA and *H. typhlonius* netMFI ratio was below 100.

DNA was extracted from mouse samples (n = 241) including liver (n = 16), duodenum (n = 16) and pooled fecal samples (n = 209) of colony and sentinel mice from routine health monitoring at the DKFZ animal facility and applied to both assays. Of all included negative PCR and hybridization controls (n = 208), one was positive for *H. typhlonius* and *H. cinaedi* (0.5%) with a low netMFI value.

All 241 mouse samples had a good DNA quality control, meaning either polA (n = 237) and/or any *Helicobacter*/*Campylobacter* (n = 191) was detected. The *Helicobacter* multiplex DNA finder detected *Helicobacter* spp. in 191 samples (79%) with 93 single infections with either *H. hepaticus* as the most prevalent type, followed by *H. typhlonius* and *H. bilis* (Figure 1). The *Helicobacter* genus was detected in all but two of the 191 samples, which were species-positive only.

Fifty samples were concordantly *Helicobacter* spp. and/or genus-negative by the conventional PCR and the *Helicobacter* multiplex DNA finder and 189 were concordantly positive for any *Helicobacter* spp. and/or genus, resulting in a kappa of 0.98 (95%CI 0.94–1.00).

Of all 189 concordantly *Helicobacter*-positive samples, 48 samples were excluded from further analyses since conventional PCR detected *Helicobacter* genus only without subsequent species-specific analysis.

Among these 141 samples, the overall species concordance was substantial (kappa = 0.614 (95%CI 0.51–0.718)) with 185 concordantly and 41 discordantly *H. bilis*, *H. hepaticus* and *H. typhlonius* identified infections (Figure 1). In total, 66 samples were identified as single infections by both assays. In one sample the conventional PCR detected *H. hepaticus*, whereas the multiplex DNA finder detected *H. typhlonius*. This one sample was negative for the genus *Helicobacter* in the multiplex DNA finder, indicating low DNA quality. Twenty-seven samples were defined as multiple-infected by the *Helicobacter* multiplex DNA finder, with 11 *H. typhlonius*, eight *H. bilis*, and 12 *H. hepaticus* identified as co-infections that were missed by conventional PCR. Three samples were identified as multiple-infected by conventional PCR only, where the additional *H. typhlonius* co-infection was missed by the *Helicobacter* multiplex DNA in all three samples. The high number of multiple-infected samples detected by the *Helicobacter* multiplex DNA finder can be explained by a higher analytical sensitivity in comparison to conventional PCR.

### 3.4. Comparison of H. hepaticus and H. bilis Multiplex Serology to Helicobacter Multiplex DNA Finder Results

In total, sera of n = 338 mice from the health monitoring of animals housed at DKFZ were analyzed for antibody responses to each of the seven antigens of *H. hepaticus* and *H. bilis*, as well as 13 *H. pylori* proteins. Four serum samples showed invalid serology results and were excluded from further analysis (Appendix A). Among the remaining 334 sera, we compared continuous as well as binary sero-responses between *Helicobacter*-negative mice (n = 57) as well as *H. bilis* (n = 26), *H. hepaticus* (n = 86), and *H. typhlonius* (n = 23) single DNA-positive mice to assess sensitivity and species-specificity of *Helicobacter* multiplex serology.

Continuous antibody responses (in MFI) to all *H. hepaticus* proteins except HH0407 were significantly higher in the *H. hepaticus* single DNA-positive group compared to the *Helicobacter* genus DNA-negative group. However, for proteins HH0713, HH0966, and HH1201 only, these antibody responses did not show cross-reactive responses in *H. bilis* or *H. typhlonius* DNA-positive mice (Figure 2A).

In the case of *H. bilis* proteins, only antibody responses to P167C and D were significantly higher in the *H. bilis* DNA-positive mice compared to DNA-negatives. Antibody responses to P167D were also found to be significantly higher in the *H. typhlonius* DNA-positive group (Figure 2B). Of note, we found elevated antibody responses to *H. pylori* proteins HP0875, HP0887_2 and HP1098 in *H. typhlonius* single DNA-positive mice compared to DNA-negative mice (Appendix A).

We applied a cut-off for sero-positivity to *H. hepaticus* and *H. bilis* antigens that allowed for a maximum of 5% sero-positive mice in the DNA-negative group (Table 5). The highest sero-prevalence among *H. hepaticus* DNA single-positive mice was given by *H. hepaticus* proteins HH0435 (71%) and HH0713 (64%). However, 44% of *H. typhlonius* and 35% of *H. bilis* single-positive mice were also sero-positive to HH0435 (Table 5). HH0713, in contrast, appeared more specific, with only 4% sero-positives in *H. typhlonius* and *H. bilis* DNA-positive mice. The highest sero-prevalence among *H. bilis* DNA single-positive mice was achieved with *H. bilis* proteins P167C and P167D (46% and 62%, respectively). Sero-prevalence to these two proteins was low in *H. hepaticus* (P167C: 11%, P167D: 13%) and *H. typhlonius* (P167C: 17%, P167D: 22%) single-positive mice (Table 5).

### 3.5. Experimental Infection of Mice with Helicobacter spp.

Each group of five mice was co-housed with a donor mouse with known *Helicobacter* infection status (*H. hepaticus* single-infected, *H. typhlonius* single-infected, dual *H. hepaticus*/*H. typhlonius infected,* and dual *H. bilis*/*H. typhlonius* infection). The expected *Helicobacter* spp. infection status of all donor mice was confirmed by the *Helicobacter* multiplex DNA finder and remained positive at wpe 0, 2 and 16. In total, 178 of 180 (98.9%) bi-weekly collected fecal samples of exposed mice had a valid DNA quality. *H. typhlonius* DNA was identified two wpe in all five single-infected mice and in nine out of ten mice with dual infection. *H. typhlonius* DNA-positivity was detected until 16 wpe in single-infected mice and in mice dual-infected with *H. hepaticus*. However, in mice dual-infected with *H. bilis*, *H. typhlonius* DNA was not detectable after six wpe (Figure 3). *H. hepaticus* DNA was also detected two wpe and remained positive until 16 wpe in nine out of ten mice, regardless of single or dual infection. Time between exposure and DNA detection was longer for *H. bilis*, at six and eight wpe for all five mice, and was not cleared until 16 wpe.

Three negative control mice co-housed with a *Helicobacter*-negative mouse were sampled bi-weekly (n = 27 samples), and had a good DNA quality and were negative for *Helicobacter* genus but positive for *H. hepaticus* (n = 3) and *H. typhlonius* (n = 1).

The experimentally infected mice were followed-up bi-weekly for the development of antibody responses to *H. bilis* and *H. hepaticus* proteins with multiplex serology. In the case of *H. hepaticus* and *H. typhlonius* single- and dual-infection, the animals developed an antibody response to proteins HH0435 and HH0713 at the earliest four wpe and two weeks after DNA detection in feces. *H. hepaticus* single-infected mice reached a median plateau of ~8000 and ~6000 MFI to antigens HH0435 and HH0713 eight and 14 wpe, respectively (Figure 3A). The *H. typhlonius* single-infected mice developed cross-reactive antibody responses to these two proteins, albeit with an overall lower maximum median antibody response (~3500 MFI to HH0435 and ~350 MFI to HH0713) (Figure 3B). *H. hepaticus* and *H. typhlonius* dual-infected mice reached a median antibody response of ~6000 MFI and ~3500 MFI to HH0435 and HH0713, respectively (Figure 3C). None of these mice developed antibody responses to *H. bilis* proteins P167C and D.

Mice in the *H. bilis* and *H. typhlonius* dual-infected group developed antibody responses to HH0435 four wpe and two weeks after the first *H. typhlonius* DNA detection in feces (Figure 3D). At the earliest, six wpe but also two weeks after *H. bilis* DNA detection in feces, mice also developed antibody responses to *H. bilis* proteins P167C and P167D with up to 10,000 MFI at wpe 14 (Figure 3D).

A fifth group of mice was exposed to *Helicobacter*-negative mice as a negative control. None of these mice sero-converted to any *H. hepaticus* or *H. bilis* protein during the course of follow-up.

## 4. Discussion

Experimental and natural infections of mice with rodent *Helicobacter* spp. such as *H. bilis* and *H. hepaticus* are associated with inflammatory and cancerous diseases of the enterohepatic tract [10,12,13,35,36]. Since the infection can alter experimental results, health monitoring of laboratory animals detecting *Helicobacter* is recommended by the FELASA [15,37].

So far, *Helicobacter* detection in routine health monitoring is mostly accomplished by PCR covering single or a small groups of *Helicobacter* spp. and subsequent gel electrophoresis or qPCR, both being time-, labor- and/or cost-intensive [2,17,18,19,38,39,40]. Moreover, these PCR assays most often target 16S rRNA, which is known to have homologies between *Helicobacter* spp. >94%, hindering a species-specific detection [16,41]. Hence, we developed a fluorescent bead-based species-specific multiplex PCR as well as multiplex serology assay with the potential to detect up to 100 agents per reaction (according to Luminex, Austin, TX, USA). The 96-well format allows fast, simple and highly reproducible analyses of up to 500 samples in less than five days, excluding DNA extraction and data output, and offers objective identification of agents [29,42].

The *Helicobacter* multiplex DNA finder includes not only the FELASA-recommended species *H. hepaticus*, *H. bilis,* and *H. typhlonius,* but also closely related rodent- and/or human-infecting *H. pylori*, *H. heilmannii*, *H. cinaedi, H. pullorum,* and *H. muridarum*. We furthermore enclosed one distantly related *C. jejuni* as a specificity control. Additionally, the detection of the *Helicobacter* genus is included in the *Helicobacter* multiplex DNA finder, giving the opportunity to identify any *Helicobacter* besides the above-mentioned species. Additional species-specific primers and probes targeting other *Helicobacter* spp. can be integrated into the *Helicobacter* multiplex DNA finder whenever needed. Moreover, the detection of other agents such as DNA and RNA viruses, bacteria and parasites relevant for health monitoring of laboratory animals can be assimilated [29]. The integration of a DNA quality control into the *Helicobacter* multiplex DNA finder gives information about the validity of samples and reduces the number of false-negative results. Analytical sensitivity of the *Helicobacter* DNA multiplex finder ranged from 10 to 1000 copies which is comparable to published qPCR assays and to other multiplex *Helicobacter* PCR assays, assuming that one bacterial cell contains at least one 16S rRNA copy [38,39,40]. We determined the sensitivity in detecting *Helicobacter* infections in comparison to published conventional singleplex PCR assays routinely applied for monitoring laboratory animals at DKFZ, and found the novel *Helicobacter* multiplex DNA finder to be highly comparable (kappa = 0.95 (95%CI 0.901–0.999)), with a more sensitive detection of *H. hepaticus, H. bilis*, and *H. typhlonius*, analyzing 241 samples with both assays.

So far, serological assays for the detection of *Helicobacter* spp. are not routinely applied in the health monitoring of laboratory animals. Previous attempts to detect antibodies to different *Helicobacter* spp. often applied whole or membrane protein extracts that may result in the detection of cross-reacting antibodies and consequently low specificity. The application of individual proteins might increase specificity and a multiplex approach could thereby allow high sensitivity to be maintained [25,26,43]. Based on our experience from a highly sensitive and specific *H. pylori* multiplex serology, we attempted here to re-assess the performance of serology in the routine health monitoring of laboratory animals [27]. To do so, we selected a set of potential immunogenic proteins for each *H. hepaticus* and *H. bilis* and applied these in a multiplex serology assay. Comparing antibody responses to results from the *Helicobacter* multiplex DNA finder in murine samples from routine health monitoring showed, however, a low sensitivity (maximum 71% for *H. hepaticus* and 62% for *H. bilis*) of the newly developed serological assay. Additionally, we detected a substantial cross-reactivity with *H. typhlonius*-infected mice, resulting also in low species-specificity.

The low sensitivity of the antibody assay in comparison to DNA detection in fecal samples likely results from the differential ability of the infected mice to build an antibody immune response due to their genetic background and genetic modification and/or the timing of sampling in relation to time-point of infection. Using experimentally infected CD1 outbred mice, we were able to show that *Helicobacter* antibodies are indeed detectable two weeks after *Helicobacter* DNA is identified, and that a plateau of maximum antibody response was observable six weeks after DNA detection. However, this infection experiment of immune-competent mice and sampling at defined time-points after infection does not reflect real-life scenarios in routine health monitoring.

In summary, the serological assay developed here is inferior to the *Helicobacter* multiplex DNA finder when it comes to sensitivity and species-specificity. However, due to low-costs and high-throughput application in combination with screening for other serologically assessed infectious agents [44] the multiplex *Helicobacter* serology could give the opportunity to monitor more frequently than the FELASA-recommended quarterly diagnostics. A possible screening scenario could also be to run the *Helicobacter* multiplex DNA finder as triage test for serologically positive mice [45].

An additional application of interest, for both the *Helicobacter* multiplex DNA finder as well as multiplex serology, could be in human samples. In humans, *H. pylori* was classified as a carcinogen for the development of gastric cancer in 1994, and since then multiple biomarker studies have been conducted to identify individuals at risk for developing cancer [46]. A causal involvement of *Helicobacter* spp. other than *H. pylori* in human disease is less well described. *H. hepaticus and H. bilis* have also been isolated from the human enterohepatic tract, but their prevalence and specific disease associations in humans remain unclear [47,48,49,50,51,52,53]. The *Helicobacter* multiplex serology described here has already been applied in a study assessing the association of different *Helicobacter* spp. with non-alcoholic fatty liver disease, a precursor of liver cancer, and indeed sero-positivity to *H. hepaticus* protein HH0713 was associated with disease [54]. Both the established novel *Helicobacter* multiplex DNA finder and the *Helicobacter* multiplex serology can in future be applied to large epidemiological studies with human samples to obtain more knowledge about the prevalence of *H. hepaticus* and *H. bilis* and their potential association with cancer. However, DNA extraction methods might be different in human samples and could require the establishment of new protocols.

## 5. Conclusions

We have developed two multiplex assays identifying infection with different *Helicobacter* spp. by the detection of DNA in the *Helicobacter* multiplex DNA finder and antibodies in the *Helicobacter* multiplex serology. While the *Helicobacter* multiplex serology lacks species-specificity and sensitivity, the *Helicobacter* multiplex DNA finder can be readily implemented in the routine health monitoring of laboratory animals. Due to the high-throughput applicability, however, both assays are promising tools to be used in large epidemiological studies investigating *Helicobacter* infection and disease correlation in humans.

## Figures and Tables

**Figure 1 microorganisms-11-00249-f001:**
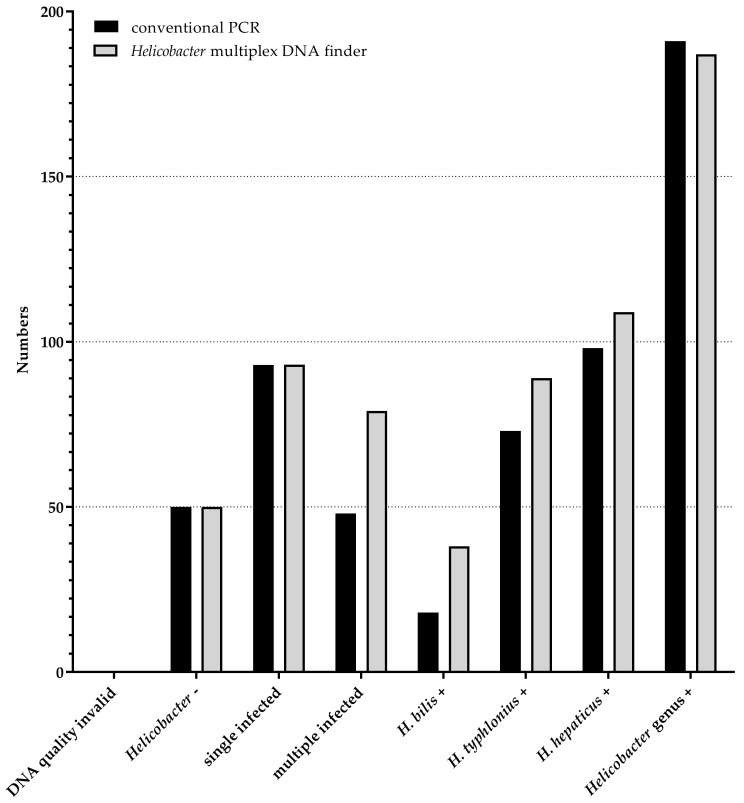
Identification of *Helicobacter* by *Helicobacter* multiplex DNA finder (grey bars) and conventional PCR (black bars). Numbers of samples defined as *Helicobacter*-negative, single- (i.e., only one *Helicobacter* spp. identified in one sample) and multiple-infected. *H. bilis*, *H. typhlonius*, *H. hepaticus*, *Helicobacter* genus positive are shown on the y-axis. The chi-squared test of independence indicated strong correlation between conventional PCR and the *Helicobacter* multiplex DNA finder (*p* < 0.0001) for any parameter compared (*Helicobacter*-, single-infected, multiple-infected, *H. bilis*, *H. typhlonius*, *H. hepaticus* and *Helicobacter* genus).

**Figure 2 microorganisms-11-00249-f002:**
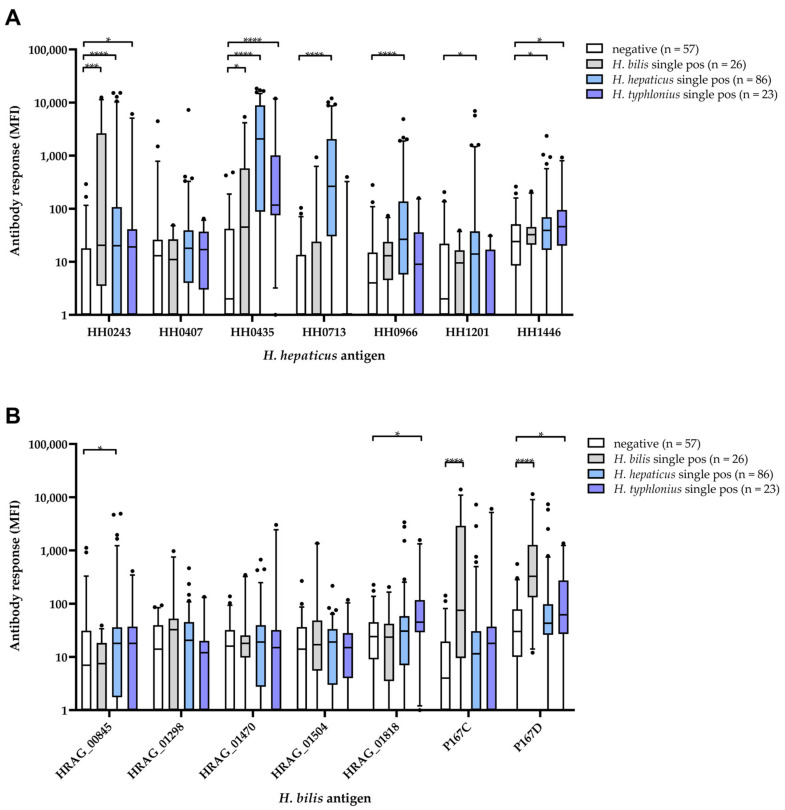
Antibody responses [MFI] to (**A**) *H. hepaticus* and (**B**) *H. bilis* antigens by *Helicobacter* multiplex DNA finder result in murine samples taken in the course of routine health monitoring. Boxes represent the 25th to 75th and whiskers the 5th to 95th percentile, solid lines show the median. Dots represent data points lying outside the 5th and 95th percentiles, respectively. Wilcoxon–Mann–Whitney test was applied to compare continuous antibody responses [MFI] in the individual DNA-positive groups to the DNA-negative group: * *p*-value < 0.05, *** *p*-value < 0.001, **** *p*-value < 0.0001.

**Figure 3 microorganisms-11-00249-f003:**
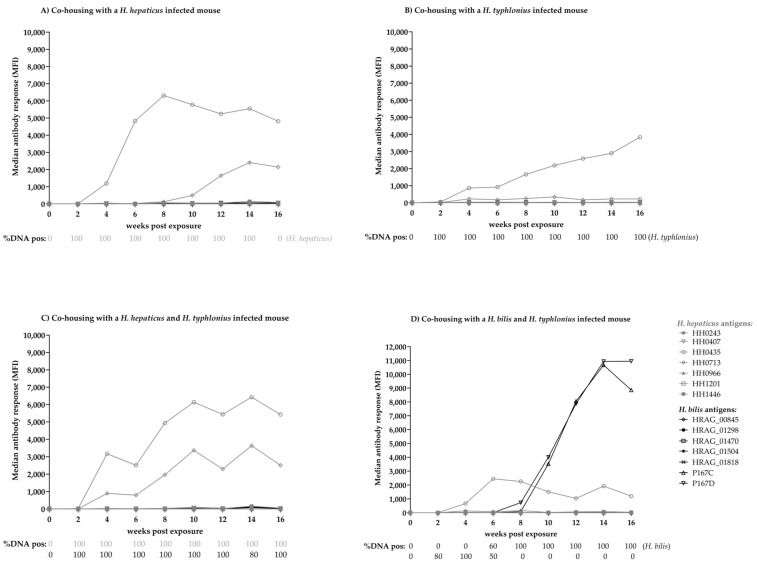
Median antibody response [MFI] to *H. hepaticus* and *H. bilis* antigens and % DNA-positivity in feces after co-housing of each group of five mice with (**A**) a *H. hepaticus*-infected mouse, (**B**) a *H. typhlonius*-infected mouse, (**C**) a *H. hepaticus*/*H. typhlonius* dual-infected mouse and (**D**) a *H. bilis*/*H. typhlonius* dual-infected mouse over a course of 16 weeks follow-up. Each group of five mice was co-housed with *Helicobacter*-infected donor mice as indicated and tested bi-weekly for DNA-shedding of *Helicobacter* in feces by *Helicobacter* multiplex DNA finder as well as for development of an antibody response to *H. hepaticus* and *H. bilis* antigens by multiplex serology. The antibody response is given as the median among the five mice per group at each time point assessed. DNA-positivity in % is given below the plot for each week post exposure.

**Table 1 microorganisms-11-00249-t001:** *Helicobacter* genus- and species-specific primers in conventional PCR and in the *Helicobacter* multiplex DNA finder.

PCR Target	Gene Target	Primer/Probe Name	Sequence (5′-3′)	Fragment Size	Reference
*Helicobacter genus*	*16S rRNA*	h276 forward	CTATGACGGGTATCCGGC	400 bp	Riley et al. [17]
		h676 reverse	ATTCCACCTACCTCTCCCA		
*H. hepaticus*	*16S rRNA*	B38 forward	GCATTTGAAACTGTTACTCTG	417 bp	Shames et al. [2]
		B39 reverse	CTGTTTTCAAGCTCCCC		
*H. typhlonius*	*16S rRNA*	Ht 184 forward	TTAAAGATATTCTAGGGGTATAT	474 bp	Franklin et al. [19]
		Ht 640 reverse	TCTCCCATACTCTAGAGTGA		
*H. bilis*	*16S rRNA*	C62 forward	AGAACTGCATTTGAAACTACTTT	638 bp	Fox et al. [18]
		C12 reverse	GGTATTGCATCTCTTTGTATGT		

**Table 2 microorganisms-11-00249-t002:** *Helicobacter* genus- and species-specific primers in the *Helicobacter* multiplex DNA finder.

PCR Target	Gene Target	Primer/Probe Name	Sequence (5′-3′)	Fragment Size
*H. muridarum*	*gyA* ^a^	fw2_murid	CCAAATGCCAGAGATGGAT	122 bp
		bw1_murid	CCGATTACATCACCAACTAT	
		p_murid	TATGAATGAGCTAAACCTTACACA	
	*ggt* ^b^	fw2_mur	GCCACTAGAGATATGTATCTTG	112 bp
		bw2_mur	GCATAGCACTCATTCCTT	
		p1_mur	AGATGTCCAATAATAGAA	
*H. pylori*	*gyA*	fw1_pylori	GATCGCYGTRGGGATGGC	144 bp
		bw1_pylori	AGTGGGAAARTCAGGCCCTT	
		p_pylori	CTTTAGYGCATGTCTTAGAA	
	*ggt*	fw1_pylori	TTAGACAAGCAAGGCAATGT	94 bp
		bw2_pylori	ACATCGCTTCCATGCCCGC	
		p1_pylori	AAATAGCCATCTTCRCTG	
*H. bilis*	*gyA*	fw1_bilis	TGCCTAATGCAAGAGATGG	126 bp
		bw1_bilis	CCATTACTTCCCCCACAAT	
		p_bilis	CGATGAAGCATAATCTAGGG	
	*ggt*	fw2_bilis	GGATAATAAAGGTAATGTGATTCC	138 bp
		bw2_bilis	GCAGGTTGCATGAGTTC	
		p3_bilis	GCTAAATATCCAAGTGTTGAAGCA	
*H. heilmannii*	*gyA*	Fw2_Hheilm	CTTGCAAATAGGCGATCT	129 bp
		bw1_Hheilm	CGCATGATCTAAGTGAAG	
		p2_Hheilm	TTCTCCTGCTCTAGCCC	
*H. typhlonius*	*gyA*	fw2_typhlonius	ATTGTAGGTAGGGCGTTA	149 bp
		bw2_typhlonius	TGGTATTTACCAATCACATC	
		p_typhlonius	GATGAACGAGCTAAGCCTTTCACC	
*H. cinaedi*	*gyA*	fw1_cinaedi	TACCAGATGCTAAAGATGG	122 bp
		bw1_cinaedi	AATCACATCGCCAACAAT	
		p_cinaedi	AATGAATGAGCTAAGCCTCTCT	
*H. hepaticus*	*gyA*	fw1_hepaticus	CCTGACGCAAAAGATGG	122 bp
		bw1_hepaticus	ATTTACCAATTACATCGCCTAC	
		p_hepaticus	AATGAATGAGCTTAATCTCTCACC	
*H. pullorum*	*gyA*	fw1_Hpull	AATGGAATAAGAGAGGCTTA	128 bp
		bw_Hpull	GCTTTATTGACCTGATAGGGA	
		p1_Hpull	TTAATGCGCCCTCTCCCTG	
*C. jejuni*	*gyA*	fw2_Cjejuni	ATGAAACTTGGTCGTTTAACA	178 bp
		bw2_Cjejuni	GAGTAATACGTGGCACA	
		p_Cjejuni	CTTGCTTGAAAATTTAATTCG	
	*ggt*	fw2_Cjejuni	TGTATCTTGATAGCAAAGGAGAA	105 bp
		bw1_ Cjejuni	GATCAAGCATAGCACTCATACC	
		p1_Cjeju	CAGCTAGATAACCTATAGT	
*Helicobacter genus*	*16S rRNA*	fw2.3_heli_uni	GAGTATGGGAGAGGTAGGTGGAATTC	110 bp
		bw2.2_heli_uni	TAATCCTGTTTGCTCCCCACGC	
		p2.2_heli_uni	CAATGAGTATTCCTCTTGA	

^a^ gyrase A, ^b^ gamma-glutamyltransferase.

**Table 3 microorganisms-11-00249-t003:** Analytical sensitivity of *Helicobacter* multiplex DNA finder.

Species	Gene	Analytical Sensitivity [# of Copies/PCR] ^a^
*H. muridarum*	*gyA*	10
	*ggt*	100
*H. pylori*	*gA*	10
	*ggt*	10
*H. bilis*	*gyA*	100
	*ggt*	10
*H. heilmannii*	*gyA*	1000
*H. typhlonius*	*gyA*	1000
*H. cinaedi*	*gyA*	10
*H. hepaticus*	*gyA*	10
*H. pullorum*	*gyA*	10
*C. jejuni*	*gyA*	10,000
	*ggt*	10
*Helicobacter* genus	*16SrRNA*	10
Mus musculus	*polA*	10
Homo sapiens	*polA*	10

^a^ determined in duplicates.

**Table 4 microorganisms-11-00249-t004:** Analytical specificity of *Helicobacter* multiplex DNA finder.

		Bacteria-Specific Probe														QC Probe
		*H. muridarum*	*H. muridarum*	*H. pylori*	*H. pylori*	*H. bilis*	*H. bilis*	*H. heilmannii*	*H. typhlonius*	*H. cinaedi*	*H. hepaticus*	*H. pullorum*	*C. jejuni*	*C. jejuni*	*Heli*	
PCR-Template	Gene	*gA*	*ggt*	*gyA*	*ggt*	*gA*	*ggt*	*gyA*	*gA*	gyA	gyA	*gyA*	*gyA*	*ggt*	*16SrRNA*	*polA*
*H. muridarum*	*gyA*	**340** ^a^	1 ^b^	1	1	1	1	1	1	1	1	1	1	1	1	1
	*ggt*	1	**124**	1	1	1	1	1	1	1	1	1	1	1	1	1
*H. pylori*	*gyA*	1	1	**30**	1	1	1	1	1	1	1	1	1	1	1	1
	*ggt*	1	1	1	**136**	1	1	1	1	4 ^c^	1	1	1	1	1	1
*H. bilis*	*gA*	1	1	1	1	**259**	1	1	1	1	1	1	1	1	1	1
	*ggt*	1	1	1	1	1	**644**	1	1	1	1	1	1	1	1	1
*H. heilmannii*	*gyA*	1	1	1	1	1	1	**87**	1	1	1	1	1	1	1	1
*H. typhlonius*	*gA*	1	1	1	1	1	1	1	**123**	1	1	1	1	1	1	1
*H. cinaedi*	*gyA*	1	1	1	1	1	1	1	5	**226**	1	1	1	1	1	1
*H. hepaticus*	*gyA*	1	1	1	1	1	1	1	1	1	**639**	1	1	1	1	1
*H. pullorum*	*gyA*	1	1	1	1	1	1	1	1	1	1	**512**	1	1	1	1
*C. jejuni*	*gyA*	1	1	1	1	1	1	1	1	1	1	1	**7**	1	1	1
	*ggt*	1	1	1	1	1	1	1	1	1	1	2	1	**92**	1	1
*Helicobacter* genus	*16SrRNA*	1	1	1	1	1	1	1	1	1	1	1	1	1	**442**	1
*M. musculus*	*polA*	1	1	1	1	1	1	1	12	1	1	62	1	1	1	**1152**
*H. canis*	whole genome	1	1	1	1	1	1	1	1	1	1	1	1	1	**610**	1
*C. lari*	whole genome	1	1	1	1	1	1	1	1	1	1	16	1	1	1	**862**
*C. coli*	whole genome	1	1	1	1	1	1	1	1	1	1	1	1	1	1	1
murine faeces	whole genome	1	1	1	1	1	1	1	10	1	1	50	1	1	1	**1095**

^a^ Signals above cut-off (value given in bold), ^b^ netMFI values of PCR products hybrized to a mixture of 15 distinct bead sets, background values were subtracted and negative values set to 1 MFI, ^c^ cross-reactivity of PCR product to non-specific probes, e.g., *H. cinaedi* PCR template with *H. pylori* probe (underlined value).

**Table 5 microorganisms-11-00249-t005:** Sero-positivity to *H. hepaticus* and *H. bilis* antigens by *Helicobacter* multiplex DNA finder result.

		N (%) by DNA Result			
Antigen	Cut-Off [MFI] ^a^	Negative (n = 57)	*H. bilis* (n = 26)	*H. hepaticus* (n = 86)	*H. typhlonius* (n = 23)
*H. hepaticus*					
HH0243	128	2 (4)	8 (31)	20 (23)	3 (13)
HH0407	956	2 (4)	0 (0)	1 (1)	0 (0)
HH0435	247	2 (4)	9 (35)	61 (71)	10 (44)
HH0713	73	1 (2)	1 (4)	55 (64)	1 (4)
HH0966	115	2 (4)	0 (0)	25 (29)	2 (9)
HH1201	136	2 (4)	0 (0)	11 (13)	0 (0)
HH1446	168	2 (4)	1 (4)	11 (13)	4 (17)
*H. bilis*					
HRAG_00845	475	2 (4)	0 (0)	4 (5)	0 (0)
HRAG_01298	84	2 (4)	4 (15)	6 (7)	2 (9)
HRAG_01470	95	2 (4)	2 (8)	8 (9)	2 (9)
HRAG_01504	89	2 (4)	4 (15)	1 (1)	1 (4)
HRAG_01818	146	2 (4)	2 (8)	6 (7)	5 (22)
P167C	89	2 (4)	12 (46)	11 (13)	4 (17)
P167D	288	2 (4)	16 (62)	9 (11)	5 (22)

^a^ cut-off applied to allow for a maximum of 5% sero-positive mice in the DNA-negative group.

## Data Availability

The data presented in this study are available on request from the corresponding author.

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
