# Peer review of "Validation of Multiplex PCR and Serology Detecting Helicobacter Species in Mice"

_microorganisms, 2023, doi:10.3390/microorganisms11020249_

Round 1
Reviewer 1 Report
The article 'Validation of multiplex PCR and serology detecting Helicobacter species in mice' describes the development of novel tests for the detection of Helicobacter species. I believe that the article extends the current wide knowledge on the species and types of Helicobacter spp. The results included in the article will allow for better and faster detection of Helicobacter spp, which can be integrated with the existing serological monitoring.
It is well referenced and overall, the manuscript is clearly written and well-structured. Additionally, the tables and figures are clear and understandable. There are some grammatical errors, however, that need to be addressed.
There are no line numbers in the manuscript, my remarks and comments will refer to pages of text and paragraphs.
AFFILIATION (1st page): Authors' full first and last names must be provided. The initials of any middle names can be added. The PubMed/MEDLINE standard format is used for affiliations: complete address information including city, zip code, state/province, and country. At least one author should be designated as the corresponding author, and their email address and other details included at the end of the affiliation section. Please edit these sections as according to MDPI guidelines.
Abstract: If the species name is written for the first time: 'Helicobacter species', please use 'Helicobacter spp' later in the text.
Helicobacter always in italics (refers to all text) in keywords, introduction, material & methods, results.
In subsection 'Helicobacer multiplex DNA finder' in 1st, 2nd and 3rd line there are no ilalics Helicobacter geneus
In a sentence ''An infection experiment was performed to assess more closely the natural history of antibody development to H. bilis and H. hepaticus in mice.'' the reference is missed.
Material and methods
In this section there are two references is missed ( of Directive 2010/63/EU, Directive 2010/63/EU,EU Directive 2010/63/EU)
Results
In section 3 (Results) Please check and add reference (7 line text)
In table 2 please correct font
Table number 3 have no siganture and in hidelines the species names have no dots (''Hmuridarum'' eg)
Please put spaces in the names of species (H.musculus eg)
Gene names should be in italics (table 3)
In figure 1 must have spaces in the names of species (H.bilis eg..)
In the table 4, use the appropriate dedicated font from the MDPI template
In section 3.5 in 2nd line 'Helicobacer' species shoud be italics and in 4th line 'Helicobacter species' the abbreviation 'spp' should be used
Disscusion
Helicobacter always in italics style (5, 27, 38, 40, 42 eg) please check this section carefully
Conclusion
In the last sentence the geneos Helicobacter should have the abbreviation spp should be used.
References
Please edit the references in accordance with the guidelines of the MDPI journal (brackets without superscript in the main text) and change the font in the list of references to the correct one.
https://www.mdpi.com/journal/microorganisms/instructions#preparation
Author Response
Dear Reviewer 1,
Thank you very much for your comments. We incorporated all your major concerns and please see for details the attached word document. Line numbers refer to the word document tracking all changes.
Kind regards,
Daniela Höfler

Reviewer 2 Report
Thank you for the chance to review "Validation of multiplex PCR and serology for detecting Helicobacter species in mice."
My opinion was that the experiment was quite laborious and time-consuming, so kudos to all of the writers for their tremendous effort.
There are concerns with the paper that must be addressed. Paper lacks line numbers for indicating points exactly.
Since differential detection of mouse-infecting Helicobacter species was described in the early 2000s, I am generally sceptical about the amount of scientific originality (e.g., Feng et al., 2005). Six to seven years ago, fluorescent hybridization probes for the detection of Helicobacter in mice models were also described (e.g., Fontenete et al., 2016). Finally, the vast bulk of references are dated (15-20 years ago). I believe that the innovation of this research (the PCR portion) might be reflected in the applicability of current cutting-edge techniques and procedures.
Several specific concerns to be addressed:
- Page 2, please explain what FELASA stands for? Also, what does 3R stands for (unclear)?
- Page 4, what is "wpe"?
- Page 5, Table 1 contains list of oligonucleotides used in the experiment. Were primers lacking specific reference generated in this study solely? If so, why this was not indicated?
- Page 6, section on Helicobacter multiplex DNA finder is rather confusing in a sense that authors failed to straightforward explain procedure. This section needs to be further clarified: "The detection of amplicons was performed via hybridization reaction, adding 10 μL of PCR products to the bead mixture. Next, heat denaturation, hybridization under stringent conditions, and incubation with streptavidin-Rphycoerythrin (Roche Diagnostics, Mannheim), followed by Luminex read-out, resulted in median fluorescence intensity (MFI) values/target for each specimen."
- Page 8, please resolve this sentence: (Error! Reference source not found.).
- Page 12, Figure 1, no indices of statistical differences displayed on bars.
This research has publication potential. In my opinion, the authors' impact would be greatly enhanced if they separated PCR and serology. The majority of the paper is devoted to the formation of antibody responses, and the innovation of the multiplex PCR finder is only mild. PCR gives identification faster and more accurately than immuno-based approaches (fact known for decades); hence, the study may be shortened and concentrated on the methodological indicators of multiplex PCR's advantage.
Author Response
Dear Reviewer 2,
Thank you very much for your comments. We incorporated all your major concerns and please see for details the attached word document. Line numbers refer to the word document tracking all changes.
Kind regards,
Daniela Höfler

Round 2
Reviewer 2 Report
I have carefully read v2 of this manuscript and appended cover letter. The authors have satisfactorily addressed most of my concerns and this revision has significantly improved the manuscript. In particular, the authors have greatly streamlined the manuscript by properly addressing ambiguities and splitting tables of concern raised.
I would recommend the manuscript for publication.